# Particulate Matter Concentration in Selected Facilities as an Indicator of Exposure to Their Service Activities

**DOI:** 10.3390/ijerph191610289

**Published:** 2022-08-18

**Authors:** Patrycja Rogula-Kopiec, Wioletta Rogula-Kozłowska, Grzegorz Majewski

**Affiliations:** 1Institute of Environmental Engineering, Polish Academy of Sciences, 34 M. Skłodowska-Curie St., 41-819 Zabrze, Poland; 2Institute of Safety Engineering, The Main School of Fire Service, 52/54 Słowackiego Street, 01-629 Warsaw, Poland; wrogula@sgsp.edu.pl; 3Institute of Environmental Engineering, Warsaw University of Life of Sciences, 166 Nowoursynowska St., 02-776 Warsaw, Poland; grzegorz_majewski@sggw.edu.pl

**Keywords:** particulate matter, indoor air, air pollution

## Abstract

In recent years, the correlation between the concentration of pollutants in the atmosphere and inside buildings has been reported as high. The air inside living quarters and public utility buildings or the interiors of public transport vehicles, as well as the relationship between the internal and external sources of particulate matter (PM) and gaseous pollutants, have underwent sufficient research. On the other hand, non-production rooms, i.e., offices, restaurants, beauty salons, etc. remain very poorly recognized in this respect. For the above reasons, the aim of this work is to determine the difference in the total dust (TSP) and respirable PM (PM4) concentrations in selected rooms, i.e., offices and beauty centers, in relation to their outdoor concentrations. They were measured at six locations in accordance with the standard for the conditions at workplaces by means of PM aspirators. Indoor concentrations of TSP and PM4 were much higher than those in the external surroundings of the facilities. There were no significant relationships between the TSP and PM4 concentrations inside and outside tested rooms. Although the characteristic of the internal PM essentially depends on the characteristics of the external PM migrating to the interior of the premises, considering some types of non-production premises, internal emissions fundamentally changed the characteristics of PM.

## 1. Introduction

Air pollution does not only have negative effects on the climate, ecosystems and human health, but more and more often is associated with financial and economic effects for entire countries or regions [1,2].

Particulate matter (PM), both atmospheric and indoor, is priority pollution, with a short-term impact on local living conditions and wellbeing and a long-term impact on human health and the environment. Virtually almost every process may involve the emission of particulate matter or its gaseous precursors to the atmosphere. PM in rooms may originate from both external (atmospheric PM) and internal sources (indoor PM).

Studies conducted in recent years show that the correlation between the concentration of pollutants in the atmosphere and inside buildings is high. It has been shown that in the absence of internal sources, the concentrations of fine PM in the indoor air are similar or higher than that in the atmospheric air [3,4,5,6,7,8,9,10,11,12]. These relationships are disturbed by internal sources of pollution and ventilation conditions. In the above-mentioned studies, the main sources of PM in rooms were identified as: human skin, hair, plants, animals, food preparation, building materials, smoking, heating (burning coal, wood and biomass), cleaning agents and activities related to living. In addition to particulate matter, internal sources also emit other pollutants, i.e., gaseous precursors of PM.

The number of papers that describe the issues of indoor air quality is very large. In the last dozen or so years, interest in research on indoor air pollution has grown significantly. For this reason, the number of publications in this subject area has increased almost several times. More and more emerging information speaks about the harmfulness of PM and its impact. The results of scientific studies appearing more often (one can observe great progress in the measurement methods) alarm that air pollution with PM in rooms and workplaces is often the same or even worse than in atmospheric air [10,13,14,15,16,17,18,19,20,21,22,23]. It is also worth noting that the air inside living quarters and public utility buildings (schools, kindergartens, theaters, museums) or the interiors of public transport vehicles, as well as the relationship between the internal and external sources of PM and gaseous pollutants and air quality, have been researched quite well so far. On the other hand, non-production rooms, i.e., offices, coffee shops, kitchens, restaurants, hairdressing salons, beauty salons, etc. remain very poorly recognized in this respect.

The aim of this work is to determine the difference in the concentrations of the total (TSP) and respirable PM (PM4-respirable fraction). PM4 is also defined as particles entering respiratory tracts and being a health threat after deposition in the gas exchange area. For the needs of the assessment of PM impact on the environment in the workplace, it is also defined in the PN-EN 481:1998 standard, set by the Interdepartmental Commission for Updating Threshold Limit Values of Harmful Agents in the Working Environment (https://www.ciop.pl/) (accessed on 13 July 2022) in selected rooms (premises, facilities)in relation to the concentrations of these PM fractions in atmospheric air.

## 2. Materials and Methods

### 2.1. Measurement Methodology—The Concept

The research carried out consisted of collecting samples of respirable (PM4) and total PM (TSP), inside and outside each of the 6 objects selected for testing: 4 beauty salons, restaurant kitchens and printing houses. For this purpose, four identical, previously calibrated aspirators were used (two aspirators with a head for PM4 and two with a TSP head), described in detail in a previous paper [22]. The measurements were scheduled for summer and early fall. This allowed us to avoid the influence of very high concentrations of PM during the heating season (winter, early spring and late autumn) on the research results. Often in the area in question, this emission masks other phenomena and impacts emissions from sources other than the municipal one [24].

Eight-hour samples of PM of the same fraction were collected in parallel inside and two outside the room. The measurements were carried out for 20 days (four working weeks) in each research facility. In total, 40 samples of PM4 and 40 samples of TSP were collected at each point.

PM samples were taken in autumn 2016. The selected measurement period—September/October—aimed at excluding the dominating emission sources in the external environment. The concentration level of the PM4 and TSP inside and outside the tested objects was determined gravimetrically. In total, 240 TSP samples and 240 PM4 samples from all the rooms were tested. 

A statistical analysis of the obtained results, including the determination of the basic statistical parameters of the measurement series and selected statistical tests, was carried out with the use of Statistica ver. 13.0.

### 2.2. Description and Characteristics of Measurement Sites and the External Environment

All the service facilities selected for this research were in the city of Bytom. It is a city located in the Upper Silesian Agglomeration (170,000 inhabitants with an area of approximately 69 km^2^). The Upper Silesian Agglomeration is in the southern part of Poland. It is a cluster of 14 industrialized cities. At the same time, it is also one of the most polluted European regions, characterized by very high concentrations of PM [24,25,26]. Over 60% of its 2 million population are employed in the service sector, a significant part of which (mainly women; [27]) is in cosmetics and the cosmetics sub-sector, whose importance for the region’s economy continues to grow. Each of the selected measurement points and its external surroundings are described below.

#### 2.2.1. Beauty Salon No. 1

Beauty salon No. 1 is located outside the city center in the Szombierki district (the distance from the city center is about 3 km) in a relatively new housing estate. On the south side, in proximity (about 25 m) to the salon, there is one of the busiest roads in Bytom. From the north, the beauty salon is surrounded by residential buildings. On this side, the facility is not exposed to the impact of heavy car traffic. There is no emission of pollutants from industry, power plants or other point sources in the immediate vicinity of the salon. There are also no inflows of emissions from heating buildings (Figure 1) (municipal emission) because all blocks of flats in the estate are equipped with central heating installations connected to the municipal network. In the salon in question, both hairdressing services and nail treatments are performed (connecting rooms). Each room is approximately 35 m^2^. The hairdressing section is equipped with a main station and a hair washing station. The part of the salon dedicated to hand care treatments is equipped with devices and materials such as a milling machine, nail files, brushes and many cosmetics containing hydrocarbons, alcohols, esters, phenols and others. Nail coatings and gels are cured under a UV lamp.

The rooms are equipped with natural gravity/natural ventilation. The inflow of outside air into the room also takes place when it is ventilated by opening the windows (made of polyvinyl chloride; a material known as cellular PVC), which does not take place until 6 p.m., after closing the premises. During the day (during services), the windows are closed. The door of salon No. 1 is opened and closed each time the client or employee enters or leaves the room and the air exchange through the door takes place only between separate rooms. During the research period, usually 4–6 people were present at the salon at the same time, i.e., one hairdresser, one beautician, 2 clients and 1–2 waiting clients.

Measurements in salon No. 1 were carried out from 1–26 September 2016 during the salon’s working hours (10.00–18.00). The PM collectors were placed in the middle of the room between two stations/adjacent rooms, approximately 3–4 m from each of them.

#### 2.2.2. Beauty Salons No. 2 and 3

Two of the four beauty salons (2 and 3) are in the very center of the city, on the first floor (3 m above street level), in the same residential building adapted to the provision of cosmetic services. In the east, in a short distance (approx. 15 m) from the salons, there is a shopping mall. Around the other sides, there are residential buildings consisting of tenement houses for residential and service purposes. The facilities are not exposed to the impact of heavy traffic because in the immediate vicinity there are only access roads to the property. There is no emission of pollutants from industry, power plants or other point sources in the immediate vicinity of the salons. However, there are impacts from emissions from heating buildings. In most tenement houses, there are heating installations in the form of stoves or story central heating installations.

The rooms intended for service activities in the facility, constituting salons 2 and 3, are generally characterized by a small space. Each of them are approximately 18 m^2^ in size and have two workplaces that are used practically all day long. PM collectors were located between these positions, not more than 2 m from each position. The rooms are equipped with natural gravity ventilation; they also ventilate by opening the windows (made of polyvinyl chloride; a material known as cellular PVC) after 6 p.m., i.e., after closing the premises. During the day (when services are provided), the windows are closed. The doors of salons 2 and 3 are opened and closed each time the client or employee enters or leaves the room, and the air exchange through the door takes place only between the rooms and the waiting room (the common room for both salons). Usually during one shift in salons 2 and 3 there was a hairdresser, a beautician, one attendant and possibly one customer waiting in the waiting room. Measurements in salons No. 2 and 3 were carried out in the period of 29 September–31 October 2016, during working hours, i.e., 10.00–18.00.

In salon 2 there are hairdressing services (cutting, staining, styling, spa, etc.) and occasional facial cosmetic services (peeling, eyebrow shaping, eyebrow and eyelash dyeing, etc.). The room is equipped with a hair dryer, a stand for washing hair and for head massages and two beds for cosmetic treatments. During their work, a hairdresser and a beautician use various agents for cosmetic treatments, hair dyeing, hair care and modeling. Here you can meet various reagents contained in cosmetics such as creams, hair and face masks, scrubs, silks, creatine, paints, perhydrol, ammonia, etc.

Salon 3 is a nail care studio (pedicure, manicure, nail extensions, hand treatments). During the workday, the beautician uses a milling machine, nail files and many cosmetics containing hydrocarbons, alcohols, esters, phenols and others. Nail coatings and gels are cured under a UV lamp. The room is equipped with a chair for clients and employees, as well as a cosmetic table.

#### 2.2.3. Salon No. 4

Salon 4 is in the middle of the Bytom-Miechowice residential district. It is located outside the city center (the distance from the city center is about 5 km) in the middle of a cluster of commercial pavilions.

The salon is surrounded by relatively new apartment blocks (1980s), playgrounds for children and green areas. The facility is out of the direct influence of traffic and industry emissions. The distance from the main road that could have a direct impact on the air quality of the salon No. 4 surroundings is about 500 m in a straight line. This distance is filled by four rows of low-rise residential buildings. Even in the winter, there is no income from heating buildings here, because all blocks are equipped with central heating installations connected to the municipal network.

The salon is equipped with the main room (approx. 50 m^2^), in which there is not only a hairdresser, but also a hand care station; in addition, there is a small room in the building which serves as a waiting room for the clients. The facility also has a social room (approx. 10 m^2^) and a toilet. Basically, everything is one whole divided by thin partitions walls. There is no possibility of opening the windows in the salon (they are glued in) and air flow is possible here thanks to gravity ventilation and opening the door.

The salon offers hairdressing services as well as hand and nail care treatments. During work time, there are usually four people in the salon at the same time (a client and an employee at each position).

The measurements in salon 4 were carried out in the period of 29 September–31 October 2016, during the salon’s working hours (10.00–18.00). Differences in the measurement periods were caused by restrictions related to the availability of the equipment. The collector was placed in the middle of the room between the two work positions: at about 2–3 m from each of them.

#### 2.2.4. Restaurant Kitchen

The professional restaurant kitchen is in a large restaurant where about 30 tables are served every day from 10.00 to 23.00. The restaurant is located outside the city center, in the Miechowice district (approx. 6 km from the center). It is located on the sidelines of housing estate development. The distance to the road, which is characterized by relatively heavy car traffic, is about 450 m west in a straight line (filled with green areas). About 30% of the surrounding buildings are heated with stoves while the rest of them are supplied from the municipal network. There are no power plants, large industrial/production plants or any large communication arteries in the immediate vicinity of the restaurant.

The restaurant is located at the ground floor of the renovated building. This building used to be a large family bakery. The restaurant kitchen is in a separate room (approx. 40 m^2^) and is connected to one of the three restaurant rooms. The kitchen is equipped with gravity ventilation and, additionally, efficient air extraction from the cooking stations. It is also equipped with two electric ovens, two gas cookers and one wood-fired oven for baking bread and pizza. There are usually 6 people in the kitchen during the working day. This is an example of a typical cuisine found in relatively modern restaurants and bars in Polish cities. There are about 50 such facilities in Bytom and over 1000 in the conurbation.

The restaurant rooms are equipped with natural ventilation and are ventilated by opening windows (made of polyvinyl chloride; a material known as cellular PVC) after closing the premises (if necessary or during hot weather, also during the provision of services). The windows are usually closed during the day. The door is opened and closed each time a customer or employee comes out of or leaves the premises.

Measurements in the restaurant kitchen were carried out during the restaurant’s working hours, in the period 1–26 September 2016, during high customer traffic, i.e., 12.00–20.00. The collectors were located about 3 m from the wood-fired oven (almost exclusively cherry).

#### 2.2.5. Printing House

The printing house selected for this research is in Bytom, in the Miechowice district (the facility is about 6 km away from the city center). The printing house is located on the sidelines of housing estate development, in the immediate vicinity of the restaurant kitchen. The facility is located about 200 m from the road with heavy car traffic. About 30–40% of the housing estate surrounding the printing house is heated with individual furnaces (hard coal, wood). The other half of the housing development is heated from the municipal network. There are no power plants, large industrial/manufacturing facilities or any major communication arteries in the vicinity of the printing facility.

In addition to the hall with printing machines with an area of about 50 m^2^, in the printing house (a new building) there is also a customer service office (about 15 m^2^), a social room (10 m^2^) and a toilet.

The building is equipped with natural ventilation. In the room where the measurements were taken (the hall), there are also two large windows (PVC) on each wall. In the summer, when the measurements were carried out, both the door between the room and the office together with the exit door (hall emergency exits) and the windows were constantly open. All prints from the machine are dried inside the hall (with natural air flow), while the machine is kept in standby mode.

The printing house provides printing services, such as large-format printing on various types of paper and foils, as well as other services, such as printing business cards, leaflets and the thermal processing (welding) of the edges of selected products (banners, posters). The sampling point was in the hall, approximately 2–3 m in a straight line from the printing machine. Sampling took place during the intensive work of the printing house, i.e., from 7.00 a.m. to 3.00 p.m., in the period 1–26 September 2016.

### 2.3. Determination of TSP and PM4 Concentrations Inside and Outside the Rooms

The concentrations of TSP and PM4 inside and outside the tested points were measured in accordance with the Polish Standard specifying the measurement conditions at workplaces. All the measurements were made with aspirators GilAir PLUS (Ekohigiena, Radom).

Aspirators consist of a pump equipped with a battery, an electronic flow rate controller and a measuring head. The measuring head for the PM4 sampling aspirator is two-stage; the first stage is a coarse dust selector and the second one includes a filter that traps all particles from the air stream directed to the second stage of the aspirator from the coarse dust selector. The air flow rate in the aspirators was 2.0 l/min for TSP and 2.2 l/min, for PM4.

The TSP and PM4 samples were collected on Whatman 25 mm diameter quartz filters (QMA, ø25 mm; GE Medical Systems, Life Sciences, Warsaw). Before and after exposure filters were conditioned in a weighing room, in a laminar chamber, at a constant temperature of 20 °C (±2 °C) and a constant air humidity of 50% (±10%), for a minimum of 48 h. Then, each filter was weighed using a Radwag balance with a resolution of 1 µm. Each time, the balance was checked and calibrated and the filters were weighed.

## 3. Results and Discussion

The average concentrations of PM during the research period in all locations were relatively high. This can be assessed by referring them, for example, to the values of the concentrations measured in the atmospheric air during tests conducted in the last decade, which were aimed at, inter alia, comparing the PM concentrations in the atmospheric air with the PM concentrations in selected offices, schools and kindergartens [10,12,28,29,30]. So far, in the scientific literature has not presented the results of air analyses in the premises of service facilities, as those considered in this paper. Therefore, for comparison, the premises that are most often tested (in terms of indoor air pollution with particulate matter) were listed, as well as those considered as workplaces for people from the service sector.

According to the WHO and EU guidelines, there are limits for PM10 and PM2.5 (PMx—particles with an aerodynamic diameter of no more than x μm) in outdoor air and for the total particulate matter (TSP) [31,32] For PM4 and TSP, the permissible concentration is determined only for selected workplaces-mainly in production halls, factories, etc. (https://isap.sejm.gov.pl/isap.nsf/download.xsp/WDU20210000325/O/D20210325.pdf, accessed on 13 June 2022). Considering the fact that in our experiment we determined mean 8 h concentrations of TSP and PM4, we cannot directly refer them to these standards. With some care, simple and very coarse comparisons can be made only for outdoor air concentrations. The arithmetic mean concentration for TSP, depending on the averaging period, was at the level of 54–170 µg/m^3^ while the permissible average daily concentration for PM10 according to the EU guidelines is 40 µg/m^3^ and the permissible average hourly concentration is 150 µg/m^3^.

Assuming that PM10 by mass is from 70–90% of TSP [33,34], it is clearly visible that the TSP concentrations in the atmospheric air in Bytom were high. Similarly, a simple comparison of the permissible concentrations of PM2.5 with the average concentrations of PM4 (by mass PM2.5 is even 90% PM4 [34]) indicates that the concentration of the respirable fraction in Bytom in the autumn period is high. With the permissible average concentration for PM2.5 at the level of 10 or 25 µg/m^3^, according to the WHO or EU guidelines, respectively, the concentrations of PM4 in Bytom reached an average of 11 to 73 µg/m^3^. It is worth noting that in the winter, when municipal emissions dominate in this region, the concentrations of PM2.5 and PM10 are often several times higher than in the autumn [35,36,37]. Thus, the very background for indoor air in Bytom seems high.

Exceptionally high concentrations of TSP in the atmospheric air, in Bytom, were recorded in the first measurement period, in September, when parallel measurements of PM concentration in the atmospheric air, as well as in the restaurant kitchen, printing house and beauty salon No. 1, were carried out. The arithmetic means for the three sets of twenty, 8 h of concentrations in this period, ranged between 156 and 169 µg/m^3^. In the second measurement period, in October, the TSP concentrations were much lower—on average, they were three times lower than the average for the September campaign (Table 1). Nevertheless, it is worth following the 8 h concentration spread in both periods, expressed by their minimum and maximum values, and the 8 h concentration spread around the arithmetic mean, expressed as the standard deviation from the mean (Table 1). The concentration range of 8 h TSP in both measurement periods was very large, as was the dispersion of the concentrations values, in relation to the mean value. In the case of a set with such different values, the median seems to be a more adequate indicator for the assessment of the average concentration of PM in the analyzed research periods [38,39].

In the case of the arithmetic mean, the average values are significantly overestimated by several measurements carried out on days characterized by relatively high wind speed (the first five measurement days; Figure 2A,C,E). In those days, the TSP concentrations were probably influenced by the secondary emission of coarse PM from the ground (soil, dust and sand resuspension) [40]. This is further proof that the concentrations of PM10 cannot be directly compared with TSP. Although, as a rule for longer averaging periods, PM10 mass constitutes the majority of TSP, there are periods such as the one we observed where particles much larger than 10 µm most likely shape the mass of TSP in the atmospheric air. Certainly, the same is also observed in indoor air in the presence of active and efficient sources of secondary emissions [41,42,43].

The results of the correlation analysis of the PM concentrations and basic meteorological parameters for the first measurement period (September) basically did not confirm the relationship between wind speed and TSP concentration (low and statistically insignificant correlation coefficients; Table 2). Nevertheless, the average wind speed on the days when the TSP concentration was very high (approx. 1.2 m/s) exceeded more than twice the average wind speed in the entire measurement period (approx. 0.5 m/s). The lack of a strong relationship between wind speed and the concentration of TSP in the atmospheric air results from the fact that in the 20-day measurement period there were only a few days when the wind speed was higher than the average. A positive relationship between the TSP concentration and wind speed was found in the second measurement period, when the day-to-day variations in wind speed were significantly greater than in the first measurement period (Table 2).

Additionally, in the case of other meteorological parameters, no statistically significant correlations with PM4 and TSP concentrations were observed. This basically confirms that the problem is a small series of analyzed data, and the fact that in the measurement period it was not only the atmospheric conditions but mainly the emission that shaped the out-door concentrations of PM4 and TSP in Bytom during the measurement period. It also seems to confirm the correctness of the selection of the measuring period.

When analyzing the median of the 8 h TSP concentrations, it should be stated that they did not differ significantly either for the location (the measurement points in the immediate vicinity of all the examined rooms) or for the measurement periods (September, October). They were even and ranged from 16 to 24 µg/m^3^. This means that in both 8 h measurement periods, the TSP concentration in the atmospheric air did not exceed these values in ten out of twenty measurements. In general, from the point of view of the standard approach to determining the average concentrations based on unit concentrations (minute, hourly or daily), adopting the median as a value characterizing a data set may be controversial. Both in the routine air quality monitoring [44] and in scientific research [33,34,35,36], the arithmetic mean is most often used for this purpose. However, in the case of the data in question, we are dealing with a small number that is very diverse in terms of the values sets of the data. Therefore, it can be concluded that, to represent the average values for the collected data sets, the median value of these sets will be better than the arithmetic mean. 

For the PM4, the concentrations measured in the atmospheric air in the immediate vicinity of the tested rooms were varied, especially in the first measurement period. While in the vicinity of the printing house and beauty salon No. 1, the concentrations were at the level of 11 µg/m^3^ and in the vicinity of the printing house they exceeded 70 µg/m^3^. In October, in the vicinity of beauty salons No. 2–4, the concentrations did not exceed 47 µg/m^3^. When analyzing the medians for the 8 h sets, it can be said that in September the average PM4 concentrations in Bytom in the three selected locations did not exceed 11 µg/m^3^, while in October they were around 30 µg/m^3^. The heating season begins in Poland in mid-October, which is related to the temperature drop. The average air temperature in the second measurement period was 7.8 °C, while in the first period it exceeded 15.3 °C.

In fact, from mid-October, a significant increase in the 8 h PM4 concentration in the atmospheric air was noticed (Figure 2H,J,L). In the first measurement period, when the concentration of PM4 in the air was measured in the vicinity of the restaurant kitchen, printing house and beauty salon No. 1, the 8 h PM4 concentrations in the air were constant (Figure 2B,D,F). The exceptions were two days in September, at the point located next to the printing house, when the concentrations of the 8 h PM4 in the air exceeded the values of 800 and 400 µg/m^3^ (Figure 2D). Considering the negative correlation coefficient for the PM4 concentrations and temperature, it can be concluded that in the second measurement period, the emissions related to the combustion of various fuels for heating purposes caused an increase in the PM4 concentrations, in the atmospheric air and in relation to the first measurement period.

Moreover, on the concentration roses of PM4, it can be noticed that the highest concentrations of PM4 were obtained when the wind came from the south, northeast and northwest (Figure 3B). In these directions (mainly in the south and northwest), there are low-rise residential buildings, partly with individual heating (coal-fired furnaces). In the first measurement period, the highest concentrations of PM4 were recorded in the northern and southern directions (Figure 3A). In the northern direction, there is the main, heavily loaded city road, leading almost the entire city (it connects Bytom with two neighboring cities). In the first measurement period, also in the case of TSP, the highest concentrations were observed in the north; in relation to this fraction also in the south direction, the concentrations were high (Figure 3C). In the southern direction, there are green areas—a large park and allotment gardens. In the case of the second measurement period, the highest TSP concentration occurred with eastern and southeastern winds (Figure 3D). These are the directions where large areas of open greenery, e.g., golf courses, are located.

The 8 h PM concentrations inside the non-production rooms under study were within a very wide range (Table 1). In the case of TSP, they ranged from 17 µg/m^3^ to 1380 µg/m^3^. On average (the median), in the research period they ranged from 43 µg/m^3^ (beauty salon No. 1) to 169 µg/m^3^ (beauty salon No. 4). In general, the TSP concentrations recorded in the restaurant kitchen, printing house and beauty salon No. 1 were lower than those in beauty salons No. 2–4. The situation was quite similar in the case of PM4. The 8 h concentrations of this PM fraction ranged from 7.23 µg/m^3^ to 821.09 µg/m^3^; on average, they ranged from 59 µg/m^3^ to 202 µg/m^3^. The highest concentrations of PM4 were recorded in beauty salons No. 2–4.

In general, the concentrations of TSP and PM4 inside beauty salons No. 2, 3 and 4, higher than inside the restaurant kitchen, printing house and beauty salon No. 1, may have resulted both from the impact of internal PM sources on these concentrations and from the accumulation of pollutants inside these facilities. It is known that the concentrations of PM in atmospheric air affect those in closed spaces. Atmospheric air can therefore be treated as a significant source of PM or/and settled dust for closed spaces. Correlations between the concentration of PM in atmospheric air and in closed spaces are disturbed by the existence of internal sources and different ventilation conditions [45,46,47,48,49,50,51,52].

In the second measurement period, the concentrations of PM4 were higher in the atmospheric air than in the first measurement period (median; Table 1). Thus, fine PM migrating to the interior of these rooms could accumulate inside them. A similar phenomenon was previously observed in the case of research conducted in classrooms, in two Polish universities and in a large sports facility, as well [17,53,54]. It is even more probable considering that, in the second measurement period, airing the rooms in the beauty salons, due to the much lower temperature of the atmospheric air, was less frequent than in the first measurement period (restaurant kitchen, printing house and beauty salon No. 1) and took place only in the hours before opening and after closing the facilities. In the first measurement period, in September, the windows in the restaurant kitchen and beauty salon No. 1 were often opened during the working hours of the facilities, while in the printing house the large window at the end of the plotter hall was open practically all the time. It has been proven that this situation may change the relationship of PM concentrations inside and outside the tested rooms in such a way that for the rooms tested, in the first period (the restaurant kitchen, printing house and beauty salon No. 1), they should be more or less aligned, and for the rooms tested in the second period (beauty salons No. 2–4), higher concentrations should characterize indoor air more than atmospheric air [55,56,57,58,59,60]. Moreover, it should be noted that beauty salon No. 1 differs from salons No. 2, 3 and 4 in terms of the size and type of treatments performed there. First, its area is about 35 m^2^, while in the other salons it does not exceed 25 m^2^. The height of each of the salons is about 2.5 m, so beauty salon No. 1 has the largest volume, which is conducive to the better dispersion of PM, as well as better ventilation [61]. Salon No. 1 is mainly used for hairdressing services and treatments, while in the other salons, the concentrations of PM in the air will also be influenced by styling and nail care treatments. This was partially confirmed by the results presented in the scientific publication by Rogula-Kopiec et al. (2019) [22].

In general, PM concentrations were much higher inside the examinant rooms than in the atmospheric air (Figure 3).

Taking into account the previously considered high concentrations of PM4 and TSP in relation to the limited one, this conclusion should be worrying. On the one hand, people living in the surveyed area, including the employees of the studied facilities, breathing heavily polluted outdoor air while at the same time being exposed to high concentrations of PM during their professional work, are almost entirely under bad air conditions. This is all the more worrying as there are reports confirming that PM inside these rooms can carry a number of toxic and carcinogenic compounds [62,63]. The problem of air quality in similar facilities is described in detail in [64,65,66]. In our work, we analyzed only the concentrations of two PM fractions; however, in terms of only this property of PM, we confirmed the importance of this problem also in Poland. In fact, it seems that the more reports of this type appear in the literature, the sooner the problem will be noticed in the international arena and it will become important for decision makers in terms of determining the conditions and standards to which such rooms should meet.

In the case of the restaurant kitchen, in the period of 20 consecutive measurement days, there were six cases where the 8 h TSP concentration was higher inside than outside; for PM4, no such case was recorded. Measurements carried out in the printing house showed that in 5 of the 20 measurement days, the 8 h TSP concentration was higher outside the room than inside and the 8 h PM4 concentration was twice as high in the atmospheric air. In beauty salon No. 1, a five times higher TSP concentration on the outside was observed; for PM4, such a case did not occur. In beauty salons No. 2 and 3, there were no cases in which the 8 h TSP concentration inside the room was lower than outside, while in beauty salon No. 4 there was one such case. Generally, it can be said that in the first measuring period, when the measurements were taken in the restaurant kitchen, printing house and beauty salon, the first few days influenced the situation observed; these were days when a very high TSP concentration occurred, most likely related to soil or road dust drift.

The TSP concentration was, on average, four times higher in beauty salons No. 2–4 and was lower or the same as the outside in the restaurant kitchen, printing house and beauty salon No. 1. However, when comparing the medians of these sets of concentrations, it can be concluded that in beauty salons No. 2–4, the TSP concentration was on average 7–8 times higher than the outside, and in the restaurant kitchen, printing house and beauty salon No. 1, it was 2–4 times higher (Table 1). For PM4, significantly higher average concentrations inside all objects were observed compared to the outside, regardless of whether the mean concentrations or the median of the concentration sets were compared, but for the mean concentrations, the difference between the internal and external concentrations was smaller. When comparing the mean sets of the indoor and outdoor/atmospheric concentrations, it can be said that the former was from 3 to 9 times higher (except for printing works). The biggest difference was in the restaurant kitchen, while the smallest was in beauty salon No. 3. In the printing house, the average concentrations inside and outside were equal. Comparing the medians of the sets, the average concentration of PM4 in the printing house was six times higher than in the atmospheric air. In the remaining facilities, the relationships of these concentrations (median yields) ranged from 4 (beauty salon No. 3) to 7 (kitchen and beauty salon No. 4). Therefore, it seems that regardless of the research conditions and the method of using the non-production rooms selected for the tests, and regardless of the characteristics of these rooms, the concentrations of PM in their interior may be, on average (median), several times higher than the PM concentration in the atmospheric air, at the same time. It remains to be considered whether the concentration of PM inside rooms depends more on the external conditions and the phenomenon of the accumulation of pollutants inside the facilities or on the emission of pollutants and thermodynamic conditions in the indoor air. Some general conclusions, however, can be drawn from the analysis of the correlation between the concentrations inside and outside the tested objects. Overall, the 8 h indoor and outdoor kitchen concentrations for both TSP and PM4 were not correlated over the measurement period. This may mean that changes in the TSP and PM4 concentrations outside this facility did not significantly affect the concentrations inside the kitchen, and thus that the internal PM sources influenced the level of air pollution in the kitchen.

For the printing house, in the case of TSP, a positive correlation of the concentrations inside and outside the room was observed. Perhaps because the large window in this room was constantly open during the measurement period, changes in the concentration of coarse PM outside the printing house influenced the changes in the concentration of coarse PM inside. A similar situation was observed in the case of beauty salons No. 1–3, although for beauty salon No. 1 the observed correlation was weak and statistically insignificant.

Table 3 shows Pearson’s linear correlation coefficients for outdoor and indoor concentrations of PM4 and TSP. No correlation was observed between the concentrations of PM4 inside and outside, no matter the type of facility. Thus, in each room there were internal sources of fine PM shaping its concentration inside. In the case of TSP, the correlation was confirmed for the outdoor and indoor concentrations for all rooms except for the restaurant kitchen and beauty salon No. 1 (Table 3).

Finally, it is worth adding that the I/O ratios of PM, i.e., the ratios of internal to external concentrations in the examined objects, in almost all cases were greater than one. In the case of PM4 for the restaurant kitchen, the I/O ratio exceeded 8.5, and in the case of the beauty salons, the I/O values were in the boundaries of 3–6. This illustrates even more clearly how much the internal sources in these objects shaped the PM concentrations. This impact causes huge increases in concentrations in relation to those in the atmospheric air, and this in turn shows how a high occupational risk related to PM inhalation may occur in the case of employees staying in their interior almost every day for many years.

## 4. Conclusions

In our paper we have shown that PM4 and TSP concentrations (both 8 h and average, during the measurement period) were generally higher in the tested facilities than in the external surroundings of the facilities. The median was determined as a reliable indicator to describe the average concentrations of the tested PM fractions in the considered non-production rooms and their external surroundings. In fact, the concentrations of TSP and PM4 inside most of the studied facilities may have resulted both from the impact of internal PM sources on these concentrations and from the accumulation of pollutants (both indoor and outdoor origin) inside these facilities. We also showed that there is a relationship between the quality and intensity of the services provided in service facilities and the concentration of PM (determined by PM emissions from internal sources). Although it is obvious, so far, for this type of object, it has not been described quantitatively in the scientific literature.

Although the characteristic of the internal PM essentially depends on the characteristics of the external PM migrating to the interior of the premises, considering some types of non-production premises, internal emissions fundamentally changed the characteristics of PM. Thus, the total daily exposure of humans who work in these premises may differ from the typical exposure of humans living in the area. Therefore, the health effects of such modified exposure cannot be predicted by only considering the dose–effect relationship (measured at the nearest monitoring station, based on the level of dust concentrations and pollutants it contains).

The results obtained in this study clearly indicate that measures should be taken to assess the air quality in typical service establishments such as restaurant kitchens, printing houses and beauty salons. It seems that a good solution in small rooms of this type will be the use of filters and air purifiers, and alternatively the use of a central air purification system. One should also remember to follow the rules of occupational hygiene and avoid introducing additional sources of pollution, e.g., burning candles, incense sticks or agents masking unpleasant odors in the room. It is also clear that the indoor air quality standards for such facilities should be introduced as soon as possible. Until now, it seemed that air quality and exposure to PM should be monitored, i.e., in large plants where process and production lines are present. Meanwhile, it is clearly visible that it should be carried out in almost all workplaces. To find a compromise between the requirements or standards for PM concentrations in different workplaces and the economy, it is necessary to investigate PM in other facilities. In addition, in further research, tests should be repeated in rooms equipped with an efficient mechanical ventilation system. It is possible that the problem presented in the paper concerns objects located in old and inadequately adapted buildings.

## Figures and Tables

**Figure 1 ijerph-19-10289-f001:**
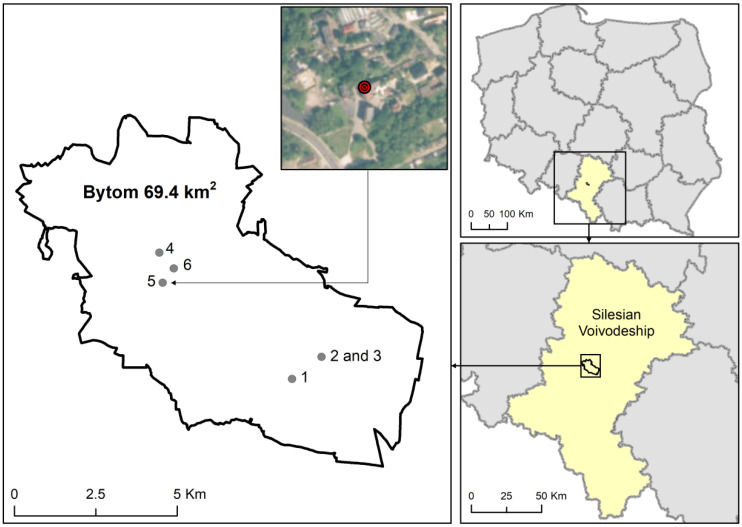
Location of the selected service facilities in the enlarged map of Bytom with a map of Poland.

**Figure 2 ijerph-19-10289-f002:**
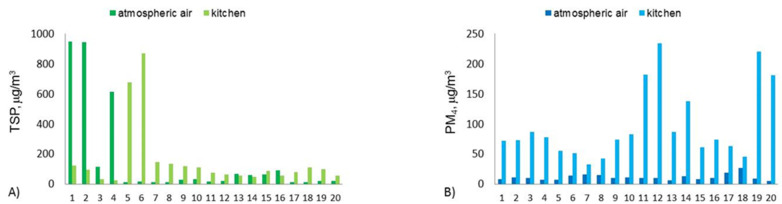
The 8 h course of TSP and PM4 concentrations inside and outside of the six studied objects in Bytom, in the period from 26 September 2016 to 31 October 2016. (**A**,**C**,**E**,**G**,**I**,**K**) show the 8-h concentrations of TSP in restaurant kitchen, printing house, beauty salon no. 1, beauty salon no.2, beauty salon no. 3 and beauty salon no. 4, respectively and in their outdoor background (atmospheric air). (**B**,**D**,**F**,**H**,**J**,**L**) show the 8-h concentrations of PM4 in restaurant kitchen, printing house, beauty salon no. 1, beauty salon no. 2, beauty salon no. 3 and beauty salon no. 4, respectively and in their outdoor background (atmospheric air).

**Figure 3 ijerph-19-10289-f003:**
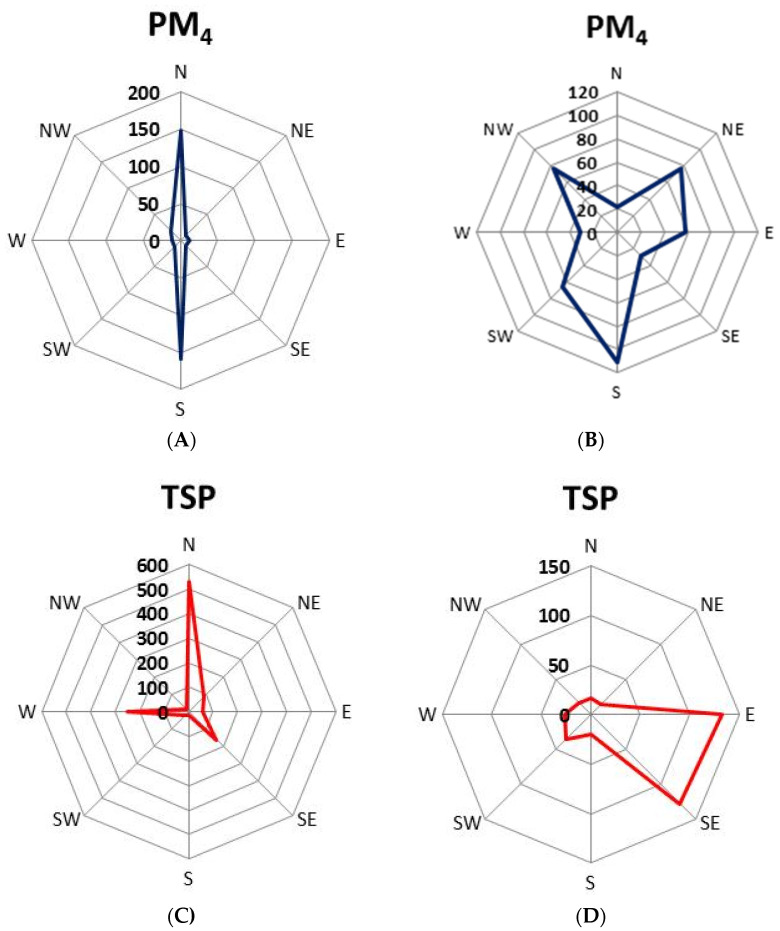
PM4 and TSP concentration roses for two measurement periods ((**A**,**C**) for the period 1–26 September; (**B**,**D**) for the period 29 September–31 October). Concentrations of PM4 and TSP are in µg/m^3^.

**Table 1 ijerph-19-10289-t001:** Descriptive statistics of the 8 h TSP and PM4 concentration sets (all values are given in µg/m^3^) measured in the restaurant kitchen, printing house, beauty salons 1–4 and in the atmospheric air, in the vicinity of these rooms, in the periods of: 1–26 September (the first three rooms) and 29 September–31 October 2016, in Bytom.

PM Fraction	Descriptive Statistics	** Restaurant Kitchen	** Printing House	** Beauty Salon No. 1	** Beauty Salon No. 2	** Beauty Salon No. 3	** Beauty Salon No. 4
Atmospheric air_TSP	minimum	11.84	11.84	11.84	5.37	2.37	5.37
	maximum	947.92	947.92	947.92	732.95	732.95	732.95
	arithmetic mean	155.83	169.27	169.09	54.42	56.76	58.01
	standard deviation	301.25	328.36	328.45	160.01	159.68	159.44
	median	24.07	23.67	23.67	16.39	18.84	21.87
Indoor air_TSP	minimum	24.62	53.50	17.52	32.75	44.11	27.70
	maximum	870.74	221.59	322.61	827.92	1383.48	773.22
	arithmetic mean	153.10	87.55	94.43	205.64	229.85	237.20
	standard deviation	217.34	46.82	107.29	208.79	286.48	179.10
	median	91.61	68.89	43.46	129.91	131.26	168.90
Atmospheric air_PM4	minimum	5.42	5.42	5.42	4.17	4.17	7.17
	maximum	26.67	841.67	25.36	183.00	255.42	255.42
	arithmetic mean	11.28	73.31	11.10	42.52	46.85	46.95
	standard deviation	4.96	204.44	4.72	44.17	57.42	56.89
	median	10.21	10.21	10.21	26.63	27.92	29.42
Indoor air_PM4	minimum	32.89	13.75	7.23	46.94	3.23	29.39
	maximum	234.38	109.58	243.82	821.09	488.87	468.66
	arithmetic mean	96.57	58.46	62.66	183.78	129.30	214.20
	standard deviation	60.37	27.69	63.41	170.06	113.96	102.55
	median	73.53	58.75	45.54	135.61	99.91	202.23

At each pair of points, the concentrations of PM4 and TSP were measured in parallel inside and outside the room, 20 times. A total of 120 values were obtained for each inside and outside PM fraction. (**) means that in the analyzed service facility (location), the differences between the TSP and PM4 concentrations (inside and outside) were statistically significant (Mann–Whitney U test; *p* < 0.05).

**Table 2 ijerph-19-10289-t002:** Pearson’s linear correlation coefficients (*p* = 0.05) of TSP and PM4 concentrations with concentrations of gaseous air pollutants and meteorological parameters.

	CO	NO	NO_2_	NOx	O_3_	SO_2_	PM10	Insolation	Air Temperature.	Air Pressure.	Wind Speed
**First Measurement Period (1 September–26 September) ^1^**											
TSP	−0.11 ^2^	−0.07	−0.16	−0.12	0.15	−0.18	−0.21	−0.00	0.12	0.14	−0.05
PM4	0.14	0.10	0.10	0.12	0.12	−0.02	0.12	0.08	0.06	0.12	−0.27
**Second Measurement Period (29 September–31 October)**											
TSP	−0.10	−0.19	−0.26	−0.22	0.13	−0.23	−0.27	−0.08	−0.01	0.03	0.51 ^2^
PM4	0.20	0.36	0.36	0.37	−0.25	0.26	0.39	−0.17	−0.19	0.46	−0.35

1 For the purposes of the statistical analysis of PM concentrations in the atmospheric air during the study period, the results of 8 h measurements were divided into two sets. The first are the results of the tests carried out in the first measurement period, i.e., practically the entire month of September, and the second are the results of the second measurement period, i.e., the measurements carried out from the end of September to the end of October. This division results from the fact that the research conducted in the first and second period concerned other areas. Printing house, kitchen and beauty salon No. 1, as well as points where PM was monitored in their external background, were located a short distance from each other, in the Bytom-Miechowice district. Beauty salons No. 2, 3 and 4, and the points of their external background, were located on the border of the city center and the Szombierki district. In each period, the tests were carried out simultaneously, at the three above-mentioned points; therefore, it was concluded that, in the case of data on PM in ambient air, they should be considered jointly for each of the measurement periods. Data on concentrations of gaseous air pollutants and meteorological conditions were taken from the station of the Provincial Inspectorate for Environmental Protection in Zabrze. The station is located approx. 5 km in a straight line from the research area, carried out in the first measurement period, and 6 km in a straight line from the test area carried out in the second measurement period. For each measuring day, hourly data from the periods when the tests were carried out inside and outside the rooms (10.00–18.00) were taken from the station and then they were averaged. ^2^ values marked in red are statistically significant (α = 0.05).

**Table 3 ijerph-19-10289-t003:** Pearson’s linear correlation coefficients (*p* = 0.05) dependence of 8 h PM concentrations in indoor air on 8 h PM concentrations in atmospheric air, for six tested objects in Bytom, in the period from 26 September to 31 October 2016.

Objects	PM4	TSP
Printing house	−0.14	0.77 *
Restaurant kitchen	−0.34	−0.15
Beauty salon No. 1	0.37	0.44
Beauty salon No. 2	−0.31	0.71
Beauty salon No. 3	−0.13	0.93
Beauty salon No. 4	−0.15	0.17

* values marked in red are statistically significant (α = 0.05).

## Data Availability

The data presented in this study are available on request from the corresponding author. The data are not publicly available due to privacy reasons.

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
