# Peer review of "Particulate Matter Concentration in Selected Facilities as an Indicator of Exposure to Their Service Activities"

_ijerph, 2022, doi:10.3390/ijerph191610289_

Round 1

Reviewer 1 Report

The article is within the scope of the journal and presents an issue of interest to the scientific society. But it must undergo some mandatory changes to be published.

I don't know PM4, what does it mean? explain in the introduction as its use is uncommon

Results

Table 1 is without the variable units. Insert!

Compare your results with your country's environmental standards and WHO standards

Multiple correlation (R2) is used to evaluate the fit of models (linear, quadratic, exponential, logarithmic and polynomials of any degree). You should use Pearson(r) correlation, which is a test whose purpose is to measure the degree of linear correlation between two quantitative variables, attribute or characteristic of a given subject. This technique is used to assess whether one variable influences the other.

Thus, the values in Table 2 will vary between 0.54 and 0.66 in PM4 to 29-09 and 31-10.

And you should interpret it like this: 0 – 0,1 = null; 0,1 - 0.3 = weak, 0.3 – 0.6 = moderate, 0.6 – 0.9 strong, >0.9 very strong correlation.

Thus, MP4 had a moderate to strong influence.

Forget (delete) the multiple correlation graphs, Figure 3, as they only detract from your article.

Recalculate Pearson correlation values (root of R2) and re-discuss results, conclusions and abstract.

So how to see your interpretation of the results was interfered by the wrong use of a statistical technique. And, yes, PM4 has an influence on the studied variable.

See these articles, as it will help in your correlation methodology and discussion about the influence on human health. Feel free to quote them.

Santana, J. C. C.; MIRANDA, A. C. ; SOUZA, L. ; YAMAMURA, C. L. K. ; COELHO, D. F. ; TAMBOURGI, Elias Basile ; BERSSANETI, F. T. ; HO, L.L. . Clean Production of biofuel from waste frying oil to reduce emissions, fuel cost, and respiratory disease hospitalisations. Sustainability, v. 13, p. 9185, 2021.

MIRANDA, A. C. ; Santana, J. C. C. ; Rosa, J.M. ; TAMBOURGI, E. B. ; HO, L.L. ; BERSSANETI, F. T. . Application of neural network to simulate the behavior of hospitalizations and their costs under the effects of various polluting gases in the city of São Paulo. Air Quality, Atmosphere & Health, v. 1, p. 1, 2021.

SANTANA, JOSÉ CARLOS CURVELO; MIRANDA, AMANDA CARVALHO ; YAMAMURA, CHARLES LINCOLN KENJI ; SILVA FILHO, SILVÉRIO CATUREBA DA ; TAMBOURGI, Elias Basile ; LEE HO, LINDA ; BERSSANETI, FERNANDO TOBAL . Effects of Air Pollution on Human Health and Costs: Current Situation in São Paulo, Brazil. Sustainability, v. 12, p. 4875, 2020

Author Response

The authors appreciate a lot all of reviewers’ and  editor’s remarks that enabled us to sufficiently revise the presentation and discussion of results and finally improve the quality of the paper. The manuscript has been revised according to the journal's requirements. The text has been formatted accordingly. Literature citations were also revised according to instructions for authors, so was the English language.

The article is within the scope of the journal and presents an issue of interest to the scientific society. But it must undergo some mandatory changes to be published.

I don't know PM4, what does it mean? explain in the introduction as its use is uncommon

Dear Reviewer, Thank you for your comments.PM4 – respirable fraction, that is the fraction penetrating to alveoli. This fraction is also defined as aerosol entering respiratory tracts and being a health threat after deposition in the gas exchange area , was defined for the needs of PM impact statement according to the PN-EN 481:1998, by the Interministerial Commission on Maximum Permissible Concentrations (MRLs) and Intensities (NDN) of Factors Harmful to Health in the Working Environment (https://www.ciop.pl/) .

Results

Table 1 is without the variable units. Insert!

Dear Reviewer, Thank you for your comments. The unit is given in the table caption - µg/m3

Compare your results with your country's environmental standards and WHO standards

Dear Reviewer, Thank you for your comments. There are limits for PM10 and PM2.5 (PMx — particles with an aerodynamic diameter of no more than x μm) in outdoor air. For total particulate matter (TSP) and for PM4 in the air, and for certain workplaces only (so-called indoor environment) (WHO 2010; Mukherjee and Agrawal 2017). The most common air pollutants exposure to humans concerns their workplace (8 hours a day). Most workplaces are not treated as dangerous. The subject of this work are concentrations of TPS and PM4 in rooms.

Multiple correlation (R2) is used to evaluate the fit of models (linear, quadratic, exponential, logarithmic and polynomials of any degree). You should use Pearson(r) correlation, which is a test whose purpose is to measure the degree of linear correlation between two quantitative variables, attribute or characteristic of a given subject. This technique is used to assess whether one variable influences the other.

Thank you for your comments. As suggested by the reviewer, we changed the calculated Spearman rank correlation coefficients to Pearson correlation coefficients – see Tables 2 and 3. The results were re-analyzed. 

Thus, the values in Table 2 will vary between 0.54 and 0.66 in PM4 to 29-09 and 31-10.

The values 29-09 and 31-10 are the measurement periods in which the results were averaged. The results were re-analyzed. 

And you should interpret it like this: 0 – 0,1 = null; 0,1 - 0.3 = weak, 0.3 – 0.6 = moderate, 0.6 – 0.9 strong, >0.9 very strong correlation. Thus, MP4 had a moderate to strong influence.

Dear Reviewer, Thank you for your comments. As suggested by the reviewer, we changed the calculated Spearman rank correlation coefficients to Pearson correlation coefficients – see Table 2. The results were re-analyzed. 

.Forget (delete) the multiple correlation graphs, Figure 3, as they only detract from your article.

Done. See Table 3.

Recalculate Pearson correlation values (root of R2) and re-discuss results, conclusions and abstract.

Done

So how to see your interpretation of the results was interfered by the wrong use of a statistical technique. And, yes, PM4 has an influence on the studied variable. See these articles, as it will help in your correlation methodology and discussion about the influence on human health. Feel free to quote them.

Done

Reviewer 2 Report

See the the pdf. document

Author Response

The authors appreciate a lot all of reviewers’ and  editor’s remarks that enabled us to sufficiently revise the presentation and discussion of results and finally improve the quality of the paper. The manuscript has been revised according to the journal's requirements. The text has been formatted accordingly. Literature citations were also revised according to instructions for authors, so was the English language.

Reviewer 3 Report

Dear authors,

As mentioned by the authors of the work, there really are no works on this subject in the literature and this is a very important point.

However, I believe that:

1 - There was no schematic drawing of the study area to understand the direction of the wind and if, at the time the sampling was carried out, the wind rose indicated the possible contribution of the emitting source or not.

2 - It would be interesting to do passive sampling in the indoor environment.

3 - Were the statistical results treated in a descriptive way, that is, were extreme results and outliers considered?

4 - It would be important, if the equipment is available, to perform energy dispersive X-ray spectroscopy on the samples collected to characterize and identify the origin of PM if they were produced in the indoor environment or came from the outdoor area.

Best regards,

Author Response

The authors appreciate a lot all of reviewers’ and  editor’s remarks that enabled us to sufficiently revise the presentation and discussion of results and finally improve the quality of the paper. The manuscript has been revised according to the journal's requirements. The text has been formatted accordingly. Literature citations were also revised according to instructions for authors, so was the English language.

1 - There was no schematic drawing of the study area to understand the direction of the wind and if, at the time the sampling was carried out, the wind rose indicated the possible contribution of the emitting source or not.

2 - It would be interesting to do passive sampling in the indoor environment.

Dear Reviewer, Thank you for your comments.

Of course, we fully agree with this remark. However, it is impossible to do the sampling only during the working time (too little material for examinations), and the costs of the analyses are very high. In addition, the work was aimed at comparing PM concentrations in different service sectors during the performance of services, i.e. during the 8-hour operating mode.

Soon, we plan to extend the research by taking samples of PAHs and BETEX to sorption tubes (by increasing the flow rate it will be possible to collect the right amount of material for analysis ).

3 - Were the statistical results treated in a descriptive way, that is, were extreme results and outliers considered?

Extreme and outliers were not removed due to:

  1. a small amount of measurement data;
  2. the impact of each of them on the average concentration - each high concentration affects the exposure, so we do not want to and cannot ignore it.

4 - It would be important, if the equipment is available, to perform energy dispersive X-ray spectroscopy on the samples collected to characterize and identify the origin of PM if they were produced in the indoor environment or came from the outdoor area.

Dear Reviewer, Thank you for your comments. Unfortunately, we do not have such equipment. Perhaps we will be able to establish cooperation on this issue and expand the research results.

Reviewer 4 Report

1.Table 2 need to revise.

2.Figure 3 need to revise clear pictures.

3.Results can list many measurement parameters explanation.

4.This manuscript can try to measure PM2.5 concentration.

5.The research results can put forward strategies and suggestions for improvement.

6.References 1, 25, and 32 are no normal cite mode that it must revise normal journal list mode.

7.This manuscript use some approach or measurement equipment method explanation.

8.Are there any calculation formulas for post-processing of measurement results?

Author Response

The authors appreciate a lot all of reviewers’ and  editor’s remarks that enabled us to sufficiently revise the presentation and discussion of results and finally improve the quality of the paper. The manuscript has been revised according to the journal's requirements. The text has been formatted accordingly. Literature citations were also revised according to instructions for authors, so was the English language.

Table 2 need to revise.

Figure 3 need to revise clear pictures.

Thank you for your comments. As suggested by the reviewer, we changed the calculated Spearman rank correlation coefficients to Pearson correlation coefficients – see Tables 2 and 3. The results were re-analyzed. 

Results can list many measurement parameters explanation.

We completed the discussion of the results

This manuscript can try to measure PM2.5 concentration.

Dear Reviewer, Thank you for your comments. In the work, the PM2.5 fraction was wilfully not measured. The smallest respirable fraction (getting into the lungs) is the PM4 fraction, which is why it has been included here. Tested PM4 fraction is in line with the Polish regulations for closed rooms and workstations

The research results can put forward strategies and suggestions for improvement.

This fragment was supplemented with paragraph conclusions

References 1, 25, and 32 are no normal cite mode that it must revise normal journal list mode.

Done

This manuscript use some approach or measurement equipment method explanation.

Both the tested PM4 fraction as well as the equipment and measurement methodology used are in line with the Polish regulations for closed rooms and workstations. This is clearly indicated in the manuscript text.

Are there any calculation formulas for post-processing of measurement results?

We do not know such formulas. It is possible that the reviewer meant to count the deposition in the lungs, etc. No studies of this type were carried out.